# Process–Based Identification of Key Tidal Creeks Influenced by Reclamation Activities

**Ying Man** [1,2], **Fangwen Zhou** [3] **and Baoshan Cui** [1,2,*]

1 State Key Joint Laboratory of Environmental Simulation and Pollution Control, School of Environment, Beijing Normal University, Beijing 100875, China
2 Yellow River Estuary Wetland Ecosystem Observation and Research Station, Ministry of Education, Dongying 257500, China
3 China Construction Eco–Environmental Group Co., Ltd., Beijing 100875, China
* Correspondence: cuibs@bnu.edu.cn; Tel.: +86-010-58802079

**Abstract:** Reclamation activities constitute a major factor threatening tidal creeks, which play an important role in the health of the ecosystem of deltas. Research on the influence of reclamation activities on the connectivity of tidal creeks is often based on changes in their morphology and ignores the process that shapes this morphology. Instead, the authors of this study focus on the influence of reclamation activities on hydrological connectivity inside the tidal creek from a process-based perspective. Changes in the hydrological distances that reflect the relative movement of sites in each tidal creek are identified and related to the resistance surface (a spatial layer that assigns values to features of the landscape, indicating the degree to which these features impede or promote movement) of the reclamation activities. We also objectively quantify the influence of different reclamation activities on the connectivity of the tidal creek. We used the proposed method to identify changes in key tidal creeks in the Yellow River Delta under the influence of reclamation activities. The results revealed the potential influence of reclamation activities (before changes appeared in the morphology of the tidal creek) from 1990 to 1995. The use of resistance surfaces thus provides a comprehensive understanding of the interactions between reclamation activities and the connectivity of tidal creeks.

**Keywords:** tidal creek; reclamation activity; hydrological connectivity

## 1. Introduction

Approximately more than 500 million people across the world live on deltas, which contribute trillions of US dollars in terms of economic revenue and ecosystem services [1,2]. Tidal creek systems play an important role in maintaining the dynamic balance of deltas. For instance, they are critical pathways to nutrients, dissolved oxygen, sediment, propagules, and the transfer of other substances between the land and the ocean [3–5]. Tidal creeks also provide a habitat and feeding grounds for fish, shellfish, and wading birds [6]. Once tidal creek systems have been disrupted, the deltas in the area become vulnerable [2,7] such that this threatens human communities in coastal areas. However, upstream and local human activities have had negative effects on the health of tidal creeks in recent decades. Reservoirs constructed upstream of them make coastal ecosystems vulnerable to reclamation activity by reducing the supply of inflowing water and sediment, and the rise in the sea level rise causes the damage from reclamation activities to become more severe [8,9]. Reclamation activities in coastal areas have received considerable attention in recent years as they may lead to a dramatic decline in regional biodiversity and ecosystem services to threaten global ecological security [10].

The influence of reclamation activities on tidal creeks is reflected in many aspects of the latter, including their morphological characteristics and changes in their connectivity [11–16], changes in water quality due to the influence of pollution [16–20], sedimentary

changes due to land use [21–23], and the responses of zooplanktons, nektons, microbenthic communities, and fish to coastal development [24–28]. Many studies have considered the above factors [6,17,29,30] and concluded that they are caused by changes in the connectivity of tidal creeks. Research on the influence of reclamation activities on the connectivity of tidal creeks often uses traditional measures of correlation. Indices and simple measures have been used to represent the degree of connectivity of tidal creeks in the presence of reclamation activities. Indices representing the connectivity of tidal creeks include structural indices based on $\alpha$, $\beta$, and $\gamma$ [31,32]. Reclamation activities have been summarized by using rough measures, such as the population density or the area of reclamation in a given region [13], in light of human activities [15]. Other indices of connectivity, such as the IC [33], use a weighted parameter to represent the degree of land use.

However, the above studies do not adequately reflect the interactions between tidal creeks and reclamation activities, particularly in a spatially explicit manner. Changes in the connectivity of a tidal creek may vary. For example, the connectivity between a given pair of sites may have significant differences from other pairs of sites because reclamation activities may occupy only one channel or part of a tidal creek. Furthermore, a variety of features of the landscape are involved in reclamation activities that might have different influences on the process and may be synergistic in some areas. To reflect this circumstance, we focus on the interaction between various reclamation activities and the hydrological connectivity between each pair of sites in each tidal creek. Reclamation activities are regarded as resistance surfaces of the landscape in the YRD. The resistance model, proposed in the context of landscape genetics, is used to quantify the resistance value of each reclamation activity by relating the observed hydrological distance (reflecting the flow of water in the system of a pair of sites in a tidal creek) to the resistance surface. This method can be used to identify the influence of human activities and elements of the natural landscape on the hydrological connectivity of tidal creeks from a process-based perspective of the landscape.

## 2. Materials and Methods

### 2.1. Study Area

The YRD (36°24′0″ N to 36°48′12″ N; 117°59′48″ E to 120°36′10″ E) is formed by the silting alluviation of the Yellow River, the second-longest river in China. Situated in the south of Bohai Bay, most of its areas are dominated by irregular semidiurnal tides [34,35]. A large area of coastal wetlands has developed in the broad tidal flats, and this makes the delta an important stopover site for the global migration of shorebirds as well as the location of coastal salt marshes. Owing to its abundant coastal resources, land reclamation activities have formed the theme of human activity in the YRD since the 1960s. Over recent decades, they have emerged as the major factor threatening the coastal environment. The main types of reclamation activities in the YRD are oil exploration, port construction, salt mining, coastal fisheries, and crop farming [36–38]. In this study, we consider the coastal wetland in the high-efficiency eco-economic zone of the YRD (Figure 1).

### 2.2. Subzone Division

As each tidal creek was constructed as a hydrological network to analyze, the range of landscapes that could cover each of them is satisfied. To reduce the amount of computation, we divided the study area into six zones according to hydrological conditions, administrative divisions, and characteristics of the reclamation activities (Figure 1). Zone 1 mainly contained the Taoer River estuary, which formed part of the course of the Yellow River in 1855. Zhanhua County is the major administrative area in this zone. Zone 2 contained the Diaokou River estuary, which formed part of the course of the Yellow River in 1976. Hekou District is its major administrative area. Zone 3 contained the Shenxian Stream estuary, which formed part of the course of the Yellow River in 1953–1976. Hekou District is the major administrative area in this zone and the Shengli Oilfield located in it. Zone 4 contained the current course, the Qingshui Stream and Qingbacha course, of the Yellow River into the sea. The National Reserve of the YRD is located in this area. Zone 5

contained the abandoned southern subdelta. Dongying District is the major administrative area and is dominated by agriculture and urban construction. Zone 6 contained Laizhou Bay. Weifang City was the major administrative area of this zone, and salt mining is the historically important industry here.

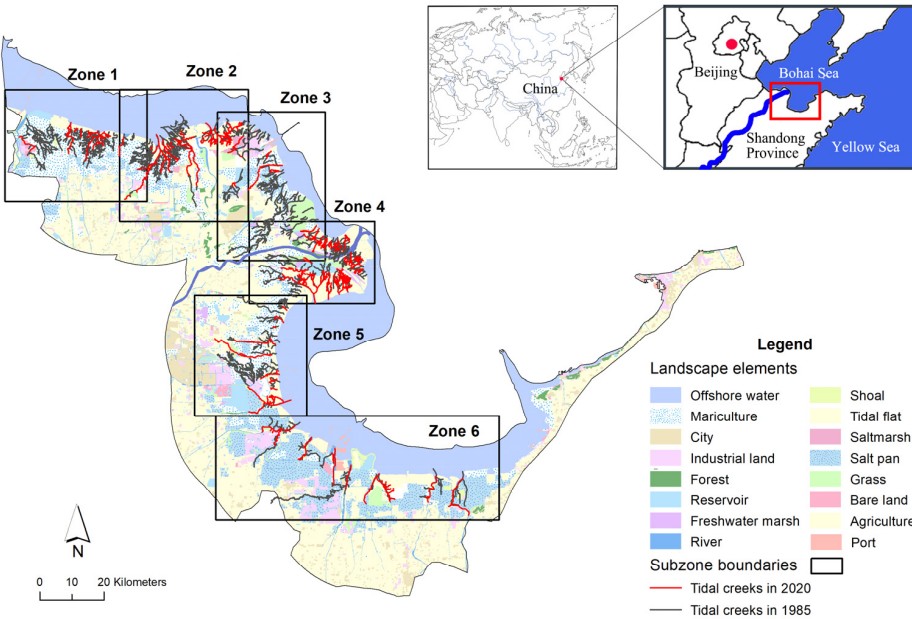

**Figure 1.** The division of the study area in the Yellow River Delta into zones, the elements of the landscape, and the tidal creeks in 1985.

### 2.3. Preprocess of Tidal Creek Hydrological Network

Information on tidal creeks in the YRD was extracted from Landsat images over the past 35 years at an interval of five years. We used the Level-1 Precision and Terrain (L1TP) data of the Landsat-5 Thematic Mapper (TM) and the Landsat-8 Operational Land Imager (OLI), which are available on the website of the United States Geological Survey (USGS; https://earthexplorer.usgs.gov/, accessed on 23 March 2022). Images with a cloud cover of less than 10% were preprocessed by ENVI 5.3 (information in the images used is provided in the Supplementary Materials Table S1). Each image was calibrated by radiometric calibration and fast line-of-sight atmospheric analysis of spectral hypercubes (FLAASH) to eliminate errors due to solar radiation, the atmosphere, and sensors. The visual interpretation of non-standard false color images was used to identify small creeks. The centerlines of the tidal creeks were extracted to construct the hydrological network.

A toolbox in ArcGIS 10.5, called the Spatial Tools for the Analysis of the River Systems (STARS), was used to construct the hydrological network of each tidal creek [39]. The nodes were classified as the source, outlet, confluence, and pseudo according to their locations in the hydrological network. The distance between each node and the outlet was calculated as the hydrological distance. We chose it to measure the flow of water between the bifurcations in each tidal creek [40] because it is the shortest distance along water channels and can better reflect the actual connections in a stream network than the Euclidean distance [41]. We then used an SSN from the R package to generate distance matrices [42]. The hydrological distance with sea reclamation activities occupation during the period of every five years among bifurcation sites was set to an infinite value (9999) to represent an infinite distance that represented the two sites were unconnected.

### 2.4. Landscape Resistance Quantification of Reclamation Activity on Tidal Creeks

According to landscape genetics, the manner in which the features of the landscape influence the flow of water in a tidal creek could be better understood by quantifying the effective distance (or connectivity) as a function of the landscape matrix [43–45]. The

latter was represented by a parameterized resistance surface, which was a spatial layer in which values are assigned to features of the landscape to represent the extent to which they facilitate or impede connectivity [44,45]. To obtain objective resistance values of the features, we used the R package ResistanceGA v.4.0-14. It uses a genetic algorithm to randomly assign values in the resistance surface [44]. By comparing and with the assistance of genetic algorithm, resistance values were assigned randomly to the resistance surfaces, and this step was taken for multiple iterations to generate different resistance surfaces. The pairwise observed hydrological distance data were compared with the resistance distances that were calculated using random-walk commute times based on each of the resistance surfaces, and likelihood was calculated using a linear mixed effect model named MLPE [44,45]:

$$Y_{ij} = \alpha + \beta(X_{ij} - x) + \tau_i + \tau_j + e_{ij}, \ j = 2, \ldots, n, \ i = 1, \ldots, j - 1 \tag{1}$$

where $Y_{ij}$ is the observed hydrological distance here, $X_{ij}$ is the resistance distance calculated based on resistance surface using circuit theory and random walk theory, $x$ is the average of resistance distances, $\alpha$ is a constant term, $\tau_i$ is a random effect that represents Y's average deviation related with location $i$ and $j$, $e_{ij}$ are assumed to be independent.

By choosing the best resistance surface with highest likelihood, the resistance value could be optimized in multiple iterations (Figure 2). Both reclamation activities and features of the natural landscape constitute the resistance surface in the YRD. The classified features of the landscape used here were taken from a dataset of Landsat images made available by Northeast Institute of Geography and Agroecology of the Chinese Academy of Sciences. We used these data to construct resistance surfaces every five years from 1985 to 2020. Each reclassified feature of the landscape was processed into raster format, and each raster cell was assigned an original resistance value (Figure 2).

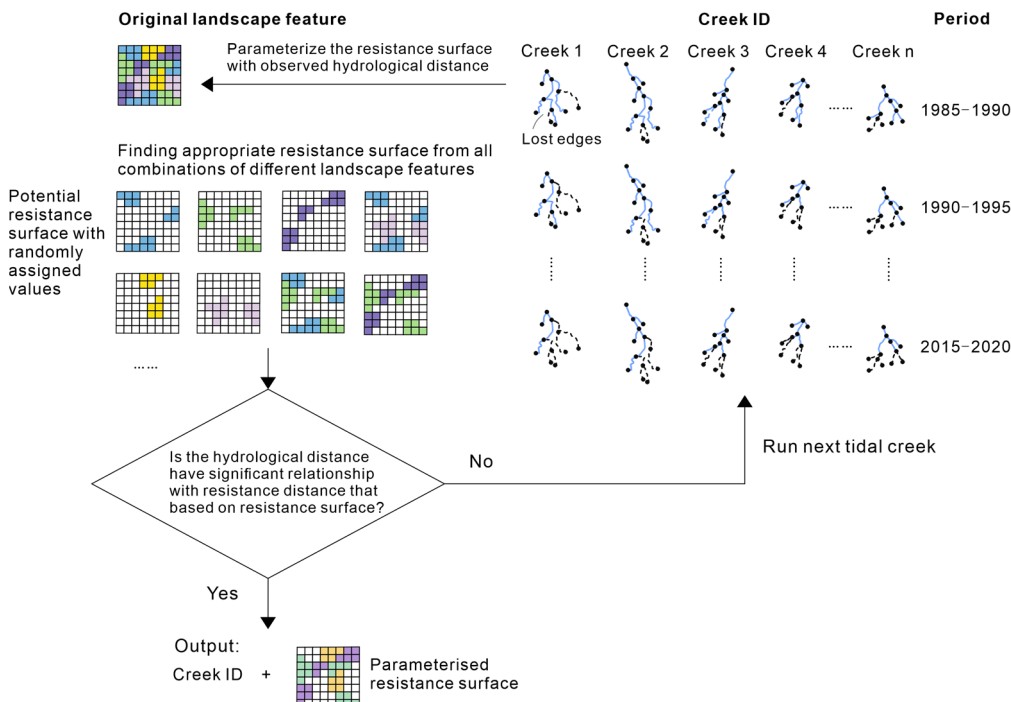

**Figure 2.** The flowchart of assessing landscape resistance surface and identification of key tidal creeks. Different colors represent different land features.

The features of reclamation activities in the YRD included urban construction, port activities, industrial work, agriculture, forestry, tourism, oilfield exploration, grassland cultivation, salt mining, rivers, reservoirs, offshore activities, and mariculture ponds. We used the similarities among the attributes of their landscapes to reclassify them into reclaimed land (land for urban construction and port activities, industrial land, agricultural land,

forests, bare land, land for tourism, and grassland), sea enclosure activities (salt panning and mariculture ponds), facilities of freshwater resources (rivers, reservoirs), and oilfield engineering (oil wells, roads to drilling platforms, and oil fields occupying a large area of land). Coastal saltmarshes, tidal flats, and offshore waters were classified as coastal wetlands of saltwater.

## 3. Results

### 3.1. Identifying Key Tidal Creeks Influenced by Reclamation Activities

The results yielded the tidal creeks influenced by reclamation activities from the perspective of landscape resistance (tidal creeks subjected to resistance-related influence, or the "influenced tidal creeks" for short) in a five-year interval from 1985 to 2020 (Table 1). The representative tidal creeks with the highest marginal $R^2$ in each period for each zone were selected. Excluding from 2015 to 2020, each period contained tidal creeks that were influenced by resistance. The from 1990 to 1995 featured the influence of reclamation activities in all zones, and this reflected the peak period of construction in the YRD. The from 2005 to 2010 featured the influence of reclamation activities on creeks in five zones (all except zone 2) and signified another peak period of construction in the YRD. Not all tidal creeks in each zone were influenced by reclamation activities in each period. The creek IDs showed that a tidal creek could be influenced by reclamation activities in different periods. For example, tidal creek 3 was significantly influenced by reclamation activities in both 2005–2010 and 2010–2015.

**Table 1.** Resistance surface optimizing results. Length represents average length of all edges in each tidal creek. $R^2$m = marginal $R^2$. Significant values are: ** < 0.01, *** < 0.001.

| Zone | Period | CreekID | Edges | Length (m) | Reclamation Activity | $R^2$m |
|---|---|---|---|---|---|---|
| 1 | 1990–1995 | 1 | 29 | 411 | Sea enclosure activity [a] | 0.74 *** |
| 1 | 2000–2005 | 2 | 10.5 | 602 | Reclaimed land [a] | 0.89 *** |
| 1 | 2005–2010 | 3 | 13.5 | 1026 | Sea enclosure activity [a] | 0.37 *** |
| 1 | 2010–2015 | 3 | 10 | 1276 | Sea enclosure activity | 0.54 *** |
| 2 | 1990–1995 | 4 | 6 | 384 | Reclaimed land [a] | 0.33 ** |
| 2 | 1995–2000 | 5 | 65 | 1057 | Freshwater resource facility | 0.14 *** |
| 3 | 1990–1995 | 6 | 18 | 795 | Freshwater resource facility; Freshwater resource facilities | 0.49 *** |
| 3 | 2000–2005 | 7 | 16.5 | 1126 | Sea enclosure activity Sea enclosure activity; Engineering in oilfield | 0.65 *** |
| 3 | 2005–2010 | 8 | 7 | 884 | Freshwater resource facility; Reclaimed land [a] | 0.90 *** |
| 4 | 1990–1995 | 9 | 7 | 1236 | Engineering in oilfield | 0.83 *** |
| 4 | 2000–2005 | 10 | 73 | 1705 | Sea enclosure activity | 0.17 *** |
| 4 | 2005–2010 | 9 | 5 | 982 | Engineering in oilfield | 0.64 *** |
| 4 | 2010–2015 | 10 | 59 | 1529 | Freshwater resource facility; Sea enclosure activity; Reclaimed land | 0.11 *** |
| 5 | 1990–1995 | 11 | 24 | 663 | Sea enclosure activity | 0.42 *** |
| 5 | 1995–2000 | 12 | 13.5 | 1904 | Freshwater resource facility [a] | 0.95 *** |
| 5 | 2000–2005 | 13 | 9 | 2129 | Sea enclosure activity | 0.33 *** |
| 5 | 2005–2010 | 12 | 15 | 1339 | Sea enclosure activity | 0.46 *** |
| 6 | 1985–1990 | 13 | 26.5 | 2144 | Freshwater resource facility; Reclaimed land | 0.13 *** |
| 6 | 1990–1995 | 13 | 26 | 2144 | Freshwater resource facility; Sea enclosure activity | 0.63 *** |
| 6 | 1995–2000 | 13 | 26 | 2144 | Freshwater resource facility; Sea enclosure activity | 0.17 *** |
| 6 | 2005–2010 | 14 | 10 | 907 | Sea enclosure activity [a] | 0.89 *** |

[a] The resistance surface contains coastal wetland of saltwater.

The scales of the tidal creeks were considerably different in terms of the number of edges and the average length. The smallest tidal creek had five edges (tidal creek 9 in zone 4 from 2005 to 2010), while the largest one had 73 edges (tidal creek 10 in zone 4 from 2000 to 2005). The longest average length was 2144 m (tidal creek 13 in zone 6) while the shortest was 384 m (tidal creek 4 in zone 2). Tidal creeks in the same zone also had different scales. The reclamation activities influencing the tidal creeks were different in terms of the level of the resistance surface (obtained by comparing the feature levels representing specific reclamation activities—for example, mariculture). Some tidal creeks were influenced by a composite resistance surface composed of multiple resistance surfaces. Examples include tidal creeks 6, 7, and 8 in zone 3, tidal creek 10 in zone 4, and tidal creek 13 in zone 6.

### 3.2. Spatial-Temporal Distribution of Landscape Resistance

We chose resistance surfaces with the highest marginal $R^2$ in each zone in the five-year period to distinguish between the distributions of the resistance values at the spatial–temporal scale (Figure 3). The results showed that high and low resistance values were distributed in different areas in different periods. Overall, the tidal creeks in the YRD were affected by both human activities and natural elements. The situation was reflected in the composite resistance surface that involved both human activities and natural phenomena. Moreover, the resistance surfaces in some zones involved only human activities, and those in others were composed solely of natural elements. There were three periods featuring resistance surfaces in zone 3 owing to the coastal wetlands of saltwater. By comparison, all resistance surfaces in zone 6 were composed of human activities.

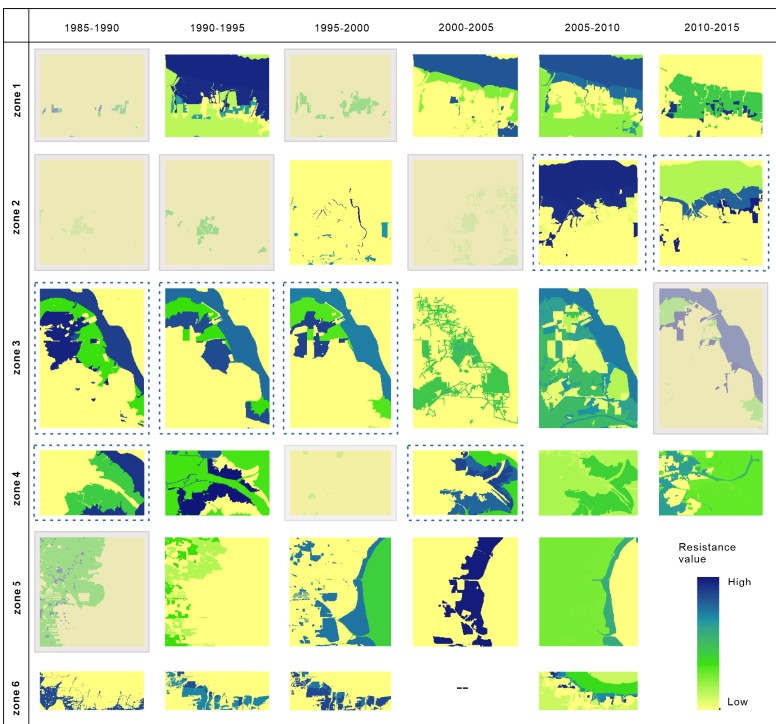

**Figure 3.** Resistance surfaces with the highest marginal $R^2$ in each zone in a five–year period. Figures with grey masks mean that the best model representing changes in the tidal creek was the distance model, followed by the resistance surface under the mask. To distinguish between resistance surfaces due to natural elements and human activities, we use dotted blue boxes to represent the resistance surfaces of the coastal wetlands of saltwater.

The tidal creeks were influenced by the resistance surfaces in zones 3, 4, and 6 in 1985–1990. Of these three zones, the resistance surfaces in only zone 6 were due to human activities. High resistance values appeared in all three zones in cases of both natural

elements and human activities. The influence of human activities on the tidal creeks appeared in four zones in 1990–1995 (zone 1, 4, 5, and 6). The highest resistance values were distributed in zones 1 and 4, and local areas of zone 6. They featured composite resistance surfaces involving both natural influence and that of human activities. The resistance surfaces involving human activities began to appear in zone 2 in 1995–2000, and the resistance values in zones 5 and 6 increased. The most influential resistance surface in zone 3 was due to natural influence (coastal wetlands of saltwater) in all three periods. The resistance surfaces involving human activities began to appear in zone 3 in 2000–2005, although with a low resistance value. The highest resistance value occurred in zone 5. Natural elements played a major role in the resistance surfaces in zones 1 and 3. The resistance values decreased in zones 4, 5, and 6 from 2005–2010. Natural elements played a more significant role in the resistance surfaces than human activities in zone 1 and 2. The resistance surfaces acted only in zones 1, 2, and 4 from 2010–2015. The resistance values were generally low, and the highest resistance values were distributed in local areas.

### 3.3. Assessing Hydrological Distances Changes

Heat maps were used to represent changes in the hydrological distances of typical tidal creeks under the influence of reclamation activities in each zone (Figure 4). The darkest shade of blue in Figure 3 represents zero hydrological distance and reflects a lost path between sites in the tidal creek.

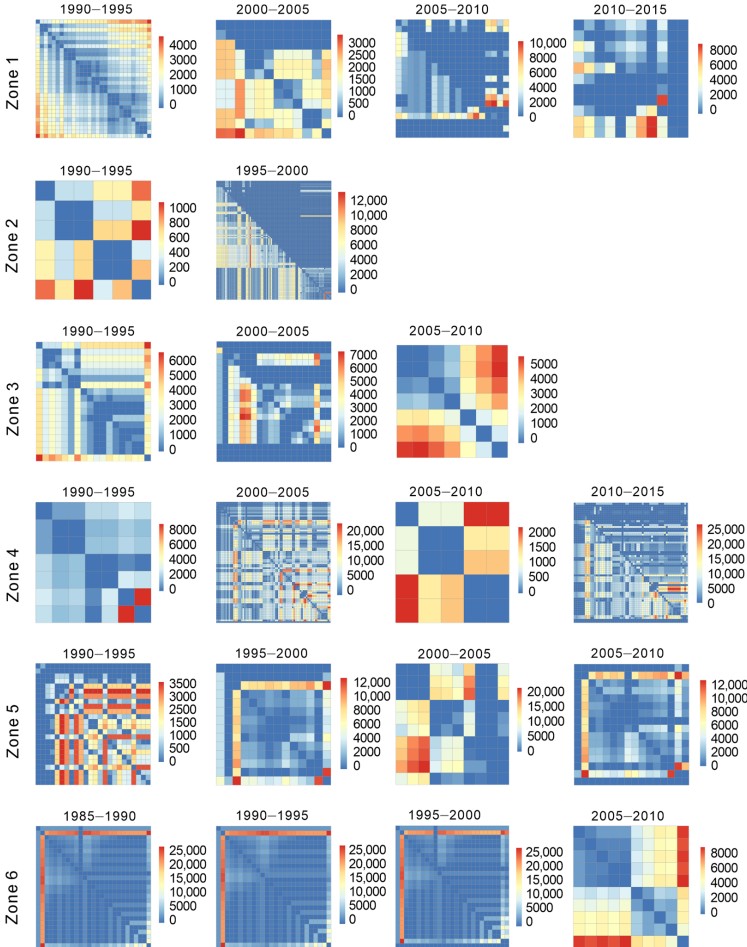

**Figure 4.** Heat maps of hydrological distances between sites in a typical tidal creek in each period. Each heat map represents a tidal creek, each column, and row represents a site in the tidal creek, and each cell represents the hydrological distance between sites in the tidal creek. The lower triangular matrix represents the beginning year of each period and the upper triangular matrix represents the end year of each period.

The hydrological distances in nearly half of the tidal creeks remained nearly unchanged over the study period. The other half exhibited prominent changes, with lost paths or sites. The tidal creeks in zone 1 from 2000 to 2015 all involved lost sites. The tidal creeks in zone 2 from 1995 to 2000 also lost sites, as did the creeks in zone 3 from 2000 to 2005, those in zone 4 from 2010 to 2015, creeks in zone 5 from 1995 to 2010, and those in zone 6 from 1985 to 1990 and 1995 to 2000. Tidal creeks lost paths in zone 1 from 2005 to 2015, in zone 4 from 2010 to 2015, and in zone 5 from 1995 to 2005. Moreover, the hydrological distances between sites decreased due to changes in the channels of the tidal creeks. Tidal creeks in zone 3 lost paths from 2000 to 2005, in zone 4 from 2010 to 2015, and zone 5 from 1995 to 2005.

By comparison with the above, new sites and paths appeared in tidal creeks in some periods. Tidal creeks with new sites were observed in zone 3 from 2000 to 2005, zone 5 from 2005 to 2010, and zone 6 from 1995 to 2000. Tidal creeks with new paths were noted in zone 1 from 2010 to 2015 and zone 5 from 2005 to 2010. Moreover, the hydrological distances between sites increased due to the appearance of new sites or paths. Tidal creeks with longer hydrological distances were observed in zone 1 from 2005 to 2015 and zone 4 from 2010 to 2015.

## 4. Discussion

The results of periods featuring reclamation activities revealed that construction peaked in the YRD from 1990 to 1995. Despite this, most tidal creeks exhibited no prominent change in this period. By comparison, another peak in construction from 2005 to 2010 led to significant changes in the form of lost sites and lost paths between sites in the tidal creeks. Many reclamation activities in the area began in 1990–1995. Examples include mariculture activities in zone 5 and the construction of a dike for the Shengli Oilfield in zone 3 [46,47]. Because the reclamation activities were in their initial stages at the time, they had only a limited direct influence on the tidal creeks. The results in this paper revealed that the reclamation activities in from 1990 to 1995 have appeared in the resistance surfaces. This indicates that although did they not have a significant direct influence on the structure of the tidal creeks, the reclamation activities affected the path of flow of water based on the resistance surfaces. Compared with the relevant studies that used the geometric properties [12,13,48] or structural indices of tidal creeks [11], our method here could identify potential threats to tidal creeks from a process-based view.

Not all tidal creeks were influenced by reclamation activities in a specific period. In other words, tidal creeks in the same area (or zone) might have been influenced by reclamation activities in different periods. This shows the heterogeneity of the influence of reclamation activities on tidal creeks. Using the proposed method while considering this phenomenon can contribute to research in the area as some previous studies have examined a small study area or a natural reserve [31,49], while others have investigated areas subjected to homogeneous reclamation activities [48,50]. Researchers can use the landscape scale method to identify threatened tidal creeks, or they can identify tidal creeks under the reclamation activities with different degrees for the first step. On this basis, they could then conduct further experiments, which would be better for making contrasts and approaching reality.

Apart from human activities, some elements of the natural landscape, such as freshwater marshes, grasslands, tidal flats, and saltmarshes, also have high resistance values. Some of the natural elements may also be under the influence of human activities, which means that the influence of human activity could also influence tidal creeks indirectly with natural elements as intermediatory. For example, saltmarshes and tidal flats may be occupied by reclamation activity. The loss of them could further influence tidal creeks as the saltmarshes and tidal flats act as resistance surfaces of tidal creeks. For saltmarsh, the relationship between saltmarshes and tidal creeks has been studied in past work [51,52]. We identified the influence of natural features of the landscape on tidal creeks in this study but did not distinguish between them in any detail. Given that saltmarshes are affected by both

human activities and climate change, this complex relationship is beyond the scope of our work here.

## 5. Conclusions

Human activities, particularly reclamation activities, have had a strong impact on tidal creeks in recent decades. The method proposed in this paper provided a quantificational analysis to assess the impact of reclamation activities on the connectivity of tidal creeks at the scale of the landscape. The key tidal creeks identified in the study were subjected to two peaks of reclamation activities. The period from 1990 to 1995 revealed their potential influence, with the reclamation activities acting as resistance surfaces. The period from 2000 to 2005 revealed the full influence of reclamation activities on tidal creeks. The resistance surfaces that influenced key tidal creeks were composed of either reclamation activities or natural features of the landscape. Reclamation activities had a major influence on tidal creeks in the later stages of the study period. The scales of the tidal creeks under the influence of reclamation activities varied considerably as well. Nearly half of the key tidal creeks lost sites or paths between sites, and new sites and paths appeared as well. This showed that reclamation activities had a varying influence on sites and paths within a given tidal creek. The work here explains the response of tidal creeks to reclamation activities in terms of their hydrological connectivity. The use of the resistance surface of the landscape to identify interactions between tidal creeks and reclamation activities provides a process-based perspective to identify the influence of human activities on the ecosystem.

**Supplementary Materials:** The following supporting information can be downloaded at: https://www.mdpi.com/article/10.3390/su15108123/s1, Table S1: Dates of satellite images used for tidal creek extraction with corresponding Landsat sensor.

**Author Contributions:** Conceptualization, Y.M. and B.C.; investigation, F.Z.; writing—original draft preparation, Y.M. All authors have read and agreed to the published version of the manuscript.

**Funding:** This research was funded by the Key Project of National Natural Science Foundation of China (U2243208), the National Natural Science Foundation of China (52271256), and the National Science Foundation for Young Scientists of China (42107057).

**Institutional Review Board Statement:** Not applicable.

**Informed Consent Statement:** Not applicable.

**Data Availability Statement:** The raw data supporting the conclusions of this article will be made available by the authors without undue reservation.

**Conflicts of Interest:** Author Fangwen Zhou was employed by the company China Construction Eco-Environmental Group Co., Ltd. The remaining authors declare that the research was conducted in the absence of any commercial or financial relationships that could be construed as a potential conflict of interest. The funders had no role in the design of the study; in the collection, analyses, or interpretation of data; in the writing of the manuscript; or in the decision to publish the results.

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
