# Peer review of "Process–Based Identification of Key Tidal Creeks Influenced by Reclamation Activities"

_sustainability, doi:10.3390/su15108123_

Round 1
Reviewer 1 Report
Manuscript ID: sustainability-2272101
Title: A process–based identification of key tidal creeks under the influence of reclamation activity
by Ying Man, Fangwen Zhou and Baoshan Cui
Written evaluation
Overall impression and recommendation
The manuscript investigates an influence of reclamation activities on tidal creeks. The paper is generally well structured, the methodology is good and results are presented clearly. There are though some minor concerns which need to be addressed as stated in comments bellow.
1) language editing required - I noticed many small grammatical errors in the paper, such as the word order, use of articles, singular and plural, and tenses of verbs. I, therefore, recommend that authors double-check their grammar, ask their native English-speaking colleague to assist to check grammar, and/or use the English editing service to ensure that it is meet the standard of publication.
2) Figure 1 – it would be maybe more visible to change subzone boundaries to black, yellow color is not the best solution; The location of the study area needs to be also presented on global scale, i.e. the exact location of study area within China or something similar
3) Table 1 – what are the ** and *** in R2m column? Where is a in superscript (noted below the table)? Please add explanation
4) Figure 2 – what happened to zone 6 from 2000-2005 and zone 5/6 from 2010-2015?
5) Lines 280-281: „Some of them 280 may also under the influence of human activities.“ I do not understand this sentence, please revise.
6) Supplementary material: Table S1 - separate the day and the month for dates, e.g. 03/14 etc.; Add explanations for TM and OLI below the table (or in table title).
Author Response
Dear reviewer,
Thank you very much for your comments and professional advice. These opinions help to improve academic rigor of our article. Based on your suggestion and request, we have made corrected modifications on the revised manuscript. Meanwhile, the manuscript had been reviewed and edited by language services of ISE (International Science Editing). We hope that our work can be improved again. Furthermore, we would like to show the details as follows:
Reviewer 1#
- language editing required - I noticed many small grammatical errors in the paper, such as the word order, use of articles, singular and plural, and tenses of verbs. I, therefore, recommend that authors double-check their grammar, ask their native English-speaking colleague to assist to check grammar, and/or use the English editing service to ensure that it is meet the standard of publication.
The author’s answer: We tried our best to improve the manuscript. The manuscript has been reviewed and edited by language services of ISE (International Science Editing), and a native English speaker was invited to help polish our article. Some changes were made to the manuscript, and these changes will not influence the content and framework of the paper. Here we did not list the changes but marked in red in the revised paper. And we hope the revised manuscript could be acceptable.
- Figure 1 – it would be maybe more visible to change subzone boundaries to black, yellow color is not the best solution; The location of the study area needs to be also presented on global scale, i.e. the exact location of study area within China or something similar
The author’s answer: Thanks for your suggestion. We have revised Figure 1. We changed the color of subzone boundaries to black and added the exact location of study area within China in the top right of the Figure 1. Hope that it is clearer.
- Table 1 – what are the ** and *** in R2m column? Where is a in superscript (noted below the table)? Please add explanation
The author’s answer: We have revised Table 1. The meaning of the **, *** in R2m column represent significant values (Specifically, *<0.05, **<0.01, ***<0.001). We’ve added his explanation in the caption of Table1. The superscript a is located in column Reclamation activity to indicate that some resistance surfaces not only contain the listed reclamation activity, but also contain coastal wetland of saltwater. And considering that the influence of natural elements on tidal creeks were not the focus in this paper (we only focused on reclamation activity), we did not list them in the table. We made a mistake indeed to forget to set the letter a in superscript conditions. Thanks for the reviewer’s remaindering, and we’ve revised Table 1. Please see page 7-8, Table1.
- Figure 2 – what happened to zone 6 from 2000-2005 and zone 5/6 from 2010-2015?
The author’s answer: There was no resistance surface model that showed significant relationship with tidal creek connectivity in zone 6 in 2000-2005 and zone 5/6 in 2010-2015 (even the distance models). Actually, the same results existed in each zone in 2015-2020, thus this period has not shown in the Figure 2 (The Figure 2 has become Figure 3 as we added a flowchart as Figure 2 in revised manuscript). The reason may come from several situations. For example, the tidal creek may change a lot during the period, in which tidal creek nodes in the end year become totally different with nodes in the starting year. The tidal creek networks we defined in this paper were based on the nodes in the starting year (e.g., hydrological nodes in 2000 were constructed as the hydrological nodes in 2000-2005) and tidal creek edges in the end year to reflect the tidal creek changes. In this situation, once the tidal creek edges changes too much that the nodes totally wander off the origin location (the location of tidal creek nodes in the starting year), the network we defined here could not be constructed. Apart from that, when null model was generated in the result, there is also no resistance surface could be used to display in Figure 2. In this situation, although the tidal creek network was existed, no resistance surface had significant relationship with tidal creek connectivity. This showed that the change of tidal creek may have no relationship with any landscape features in this temporal scale (5 year). For example, a more dramatic influence from human activity may be the reason. Therefore, as the structure of tidal creeks in the YRD become simpler after 2010, especially after 2015, it is easy to meet the two situations we listed here to cause the result that no resistance surface model existed in a 5-year temporal scale. Hope it is clearer.
5) Lines 280-281: “Some of them 280 may also under the influence of human activities.” I do not understand this sentence, please revise.
The author’s answer: We’ve revised the text, and made an explanation. We’ve changed [Some of them may also under the influence of human activities.] to [Some of the natural elements may also under the influence of human activities, which means that the influence of human activity could also influence tidal creeks indirectly with natural elements as intermediatory. For example, saltmarshes and tidal flat may be occupied by reclamation activity. The loss of them could further influence tidal creeks as the saltmarshes and tidal flat act as resistance surface of tidal creeks.] Hope that it is now clearer. Please see page 13, lines 368-373.
6) Supplementary material: Table S1 - separate the day and the month for dates, e.g. 03/14 etc.; Add explanations for TM and OLI below the table (or in table title).
The author’s answer: Thanks for the reviewer’s suggestions. We’ve revised Table S1 and explanations for TM and OLI in table caption.
Reviewer 2 Report
The authors have done a good job, but in general, I will give a final summary.
The analysis needs to be based on several factors. Climate change and complex relationships are important parameters for this research. It is precisely these factors that I miss in the study. In this form, the results and research do not correspond to the level of a journal.
After supplementing the study with a multicriteria analysis.I recommend authors to submit the article tentatively for publication.
Author Response
Dear reviewer,
We feel great thanks for your professional review work on our article. As you are concerned, the climate factors were needed to be supplemented. According to your nice suggestions, we have made a reply below.
Reviewer 2#
- The authors have done a good job, but in general, I will give a final summary.
The analysis needs to be based on several factors. Climate change and complex relationships are important parameters for this research. It is precisely these factors that I miss in the study. In this form, the results and research do not correspond to the level of a journal.
After supplementing the study with a multicriteria analysis.I recommend authors to submit the article tentatively for publication.
The authors’ answer: Thanks very much for the reviewer’s suggestions. We quite agree with the reviewer’s concern that the climate factors are important parameters. But as the paper’s title shows (Process–based identification of key tidal creeks influenced by reclamation activities), we focused only on the reclamation activities. Actually, we have collected climate data during the revised period, and we have analyzed them to some extent using the method the reviewer suggested. But in the end, the authors have come to an agreement that the climate factors may have influence on the tidal creek connectivity in complex ways which has beyond the range of this paper.
Maybe our consideration could be understood when the reviewer has known the processing of natural elements in this paper. Although we have taken natural elements into account in this paper, they were considered in a quite rough extent. Natural elements were necessary factors which act as a component of resistance surface. Once omitting the natural elements, the landscape features is not incomplete, which could influence the parameterization of resistance surface. Despite that, we did not consider the natural elements in detail. Because the interaction between natural elements and tidal creek connectivity could also be complex and the scales of the interaction may be different with the scale in this paper.
For climate factors, the situation is similar to natural elements, but with a difference. Specifically, we focused on the process-based interaction between landscape features and tidal creek connectivity, which was based on the landscape resistance model in which landscape features act as resistance in a habitat network that could be modeled as circuit network. As the movement of waterflow is based on each tidal creek, resistance surface to each tidal creek is about the area a little larger that the area the tidal creek located. Considering the length of tidal creek is at most several kilometers long, the area is about several square kilometers. In this scale, when we constructed the climate factors as resistance surface as the other landscape features did, they could not have much changes in such small area that could influence the waterflow movement of tidal creeks. And as the generalized linear mix model run and compare the likelihood of each resistance surfaces, the resistance surface of climate factor could be replaced by other resistance surfaces that may cut the tidal creeks in fact, such as some kind of reclamation activity. Apart from the spatial scale, for temporal scale, the 5-year period is also not enough to reflect the change of climate factors to influence the waterflow movement of tidal creeks. Therefore, even if we consider the climate factor, the spatial and temporal scale of this papers could not allow to reflect the influence of them. And unlike the natural elements that act as a component of the landscape features that forms resistance surfaces together with reclamation activity, the climate factor acts as independent resistance surfaces. Once their influence is less, they could be replaced by other resistance surfaces, which could be neglected. In further research, we would like to deal with climate factors when the spatial and temporal scale is appropriate. Hope this explanation could satisfy the reviewer’s concern.
Again, we appreciate for your warm work earnestly and hope that the correction will meet with approval. Once again, thank you very much for your comments and suggestions.
Reviewer 3 Report
The manuscript studies the impact of reclamation activities on tidal creek connectivity based on the process analysis method, which has certain scientific value. However, the description of the research method in the manuscript is too simple, and it is difficult for readers to understand how the subsequent results were obtained. At the same time, there is no quantitative analysis of how reclamation activities affect tidal creek connectivity, and the content is too thin. Only 1 table and 2 figures cannot support subsequent conclusions.
L128 A flowchart should be used to demonstrate the research ideas of this manuscript, and the calculation formulas of the important parameters indicators involved in the manuscript should be elaborated.
L197 How the results in Figure 2 were obtained.
L251 In the discussion section, the impact of reclamation activities on tidal creek connectivity is all qualitative analysis, and there is no quantitative research. It is not clear from the discussion how reclamation activities affect tidal creek connectivity
Author Response
Dear reviewer,
We feel great thanks for your professional review work on our article. As you are concerned, there are several problems that need to be addressed. According to your nice suggestions, we have made extensive corrections to our previous draft, the detailed corrections are listed below.
Reviewer 3#
- L128 A flowchart should be used to demonstrate the research ideas of this manuscript, and the calculation formulas of the important parameters indicators involved in the manuscript should be elaborated.
The author’s answer: Thanks for the reviewer’s suggestions. We’ve drawn a flowchart. Pease see Figure 2 in the revised manuscript. We also revised the text and added more details of the method we used including the formula. Hope that it is now clearer. Please see page 6, lines 169-184 of the revised manuscript.
- L197 How the results in Figure 2 were obtained.
The author’s answer: We showed the steps in the new drawn flowchart in Figure 2. Specifically, a R-package called Resistance GA was used to parameterize the landscape features to obtain quantified resistance values of each resistance surface. We also revised the text in section 2.4 (Landscape resistance quantification of reclamation activity on tidal creeks). Hope that it is now clearer. Please see page 5-6 of the revised manuscript, lines 160-192.
- L251 In the discussion section, the impact of reclamation activities on tidal creek connectivity is all qualitative analysis, and there is no quantitative research. It is not clear from the discussion how reclamation activities affect tidal creek connectivity
The author’s answer: In this research, we used a landscape resistance view to show the influence of human activity on tidal creek connectivity. This means that we focus on the effect of resistance from human activity in landscape scale. The landscape resistance comes from circuit theory that treated the landscape features as habitat network. Landscape features with high resistance values would impede organism’s movement, and landscape features with low resistance values would act as movement path of specific organism. Here, we treated the waterflow as the movement of organism since waterflow follows similar principle. Therefore, although the impact of human activity on tidal creek come from various aspects, we only focus on the impact from a landscape resistance view. And the quantified resistance value of the landscape features including human activities have been shown in Figure 2 of the original manuscript (Figure 2 of the revised manuscript). It is necessary to note that the value in this figure is quantified with specific value. Only that we prefer to compare the spatial and temporal difference and connection of landscape features, the resistance values were normalized and the specific values were instead of high and low colors in the legend. Thus, by observing the spatial distribution of high or low resistance value, we think it is enough to reveal the impact of human activity acting as landscape resistance on tidal creek connectivity. In addition, Table 1 also showed which human activity has led to the formation of landscape resistance and the change of tidal creeks under the impact of human activities [page 7, lines 211-214]. Hope that it is clearer.
We appreciate for your warm work earnestly and hope that the correction will meet with approval. Once again, thank you very much for your comments and suggestions.
Round 2
Reviewer 2 Report
Thank you for the addition and further explanation of the text.
Reviewer 3 Report
The author has made the revisions exactly as requested and recommends publication in this edition